# Impact of Staff Localization on Turnover: The Role of a Foreign Subsidiary CEO

**DOI:** 10.3390/bs12100402

**Published:** 2022-10-19

**Authors:** Joonghak Lee

**Affiliations:** College of Business, Gachon University, Seongnam-si 13120, Korea; joonghaklee@gachon.ac.kr

**Keywords:** staff localization, turnover, subsidiary CEO, psychological contract, South Korean MNCs

## Abstract

Great resignation has become a critical issue in management discipline and retaining talents is one of the most important properties across the globe. Among them, local staff have been regarded as an essential competitive advantage for multinational companies and their sustainability. In this sense, staff localization has received considerable attention from scholars and professionals; however, few studies have examined the mechanisms underlying the relationship between staff localization and turnover. This study examines the macro-level relationship between the ratio of local staff in a subsidiary and the actual turnover rate of 89 multinational companies in 25 countries through their headquarters and subsidiary staff. Additionally, the aim of this study was to identify the moderating impact of a CEO’s nationality. The results showed that local staff leave organizations in which there are more expatriates deployed from HQs. Furthermore, the CEO’s nationality buffered the relationship between staff localization and the local staff turnover. This study can contribute to the academia and practice by revealing the effect of staff localization on staff turnover. In addition, a CEO staffing strategy focusing on nationality can be considered an important factor in retaining competitive local staff during the COVID-19 pandemic for multinational companies.

## 1. Introduction

Multinational corporations (MNCs) are looking for ways to extend their businesses into new countries as globalisation accelerates [1]. The rapid rise in international trade and foreign investment connected with this massive economy has increased the demand for highly trained managers and professionals [2]. MNCs have hired and retained a number of local employees to maintain and expand their companies in many countries, since talented capital is increasingly vital in the ever-increasing struggle to employ the most valuable individuals in the marketplace [3,4]. Furthermore, most MNCs maintain a significant portion of their workforce, manufacturing operations, and R&D capabilities in their home country [5]. The goals of developing a global core competency, a diversity of strategic perspectives, or a multicultural frame of reference within the top management team are all factors in hiring local staff [6].

Local markets pose a considerable problem for MNCs and cultural differences, regional economic disparities, a lack of managerial expertise, and distinct customer behaviours can all be barriers to a MNC’s success in certain markets [7,8]. One solution to these problems is staff localization. As a result, some MNCs have hired local staff to fill expatriate management jobs. Most of them do not intend to increase their expatriate workforce over the next five years [9], and the number of expatriates is declining because of the high expense of keeping them overseas [10].

Although staff localization is becoming more common in MNCs, few academics have empirically examined its efficiency, aside from professional reports [8]. Furthermore, there is little evidence that increasing the use of local staff helps lower the discontent among local employees or helps to maintain competent people [2]. Therefore, studies on the relationship between staff localization and the turnover of local staff need to be further examined [8,11]. To fill this research gap, this study aimed to examine the effect of staff localization on local staff turnover in South Korean MNCs from 25 countries in three industries. In addition, since there has been little empirical research on how staff localization affects the turnover [8,10,12], this study focussed on the CEO’s nationality as a moderator in the relationship between staff localization and turnover. Although the number of expatriates is relatively high, local staff are expected to move to the CEO position if they occupy it. In that sense, the aim of this study was to first investigate the relationship between staff localization and the turnover, since there is a tendency to utilize more local staff in the subsidiaries of the MNCs to respond to the changes in the way of work affected by COVID-19 and a digital transformation. Additionally, the study sought to identify the moderating impact of a CEO’s nationality since the CEO is a strong signal to employees for career visibility. In doing so, this study contributes to the literature by revealing the impact of staff localization on turnover, especially in the 25 countries with a high generalizability. Moreover, the study used the actual data of the local staff ratio, the CEOs nationalities and the turnover rate from internal systems. In particular, previous studies have utilised the turnover intention for the independent variable due to the difficulty of gathering the actual turnover rate, which has led to a lack of research in this area. Furthermore, professionals can utilise the findings of this study by considering the impact of subsidiary CEOs as a contingency to lower the turnover of local staff. The remainder of this paper proceeds as per the following format. Hypotheses are developed based on literature reviews and guide theory. Then, the method, results, discussion, and conclusion sections follow.

## 2. Literature Review and Hypotheses Development

### 2.1. Literature Review of Subsidiary Staffing

Understanding the needs of local staff as they enter the local market is critical to corporate success in the global marketplace, especially during uncertain times [13,14,15]. Local initiatives can provide MNCs with winning prospects and a new vitality, and headquarters should lower their corporate barriers to improve feedback and to stimulate local adaptations and responsiveness from their subsidiaries [16].

The local government, local management, and a firm’s parent business were recognized as the three sources of pressure for localization [17]. Governments in several emerging-market countries have introduced work permits, expatriate quotas, and financial audits of expatriate pay to gain independence in MNC management. Moreover, as local employees gain experience, they are likely to become disgruntled with their secondary positions, as they are unable to advance their careers because of their expatriates’ management positions [18]. Finally, employers in host countries localize management positions because they are beneficial in terms of cost savings and local knowledge from the standpoint of local managers [19].

Cost savings, skill transfers, and family/personal constraints are some of the key reasons for localization in different geographic areas [20]. Expatriation costs corporations a significant amount of money, making localization a good way to minimise costs; however, if an MNC engages local staff, then the management costs can be significantly reduced. For instance, direct expatriate costs are at least three times greater than local manager costs in China [21].

Local staff are more attentive to changes in new regulations, legislation, and the environment in their country than expatriates. This also assists local staff in identifying new opportunities by adjusting strategies and processes to operational needs as well as avoiding legal blunders [7]. Expatriation also has significant disadvantages, such as frequent changes in expatriates’ managerial roles and the scarcity of well-prepared expatriate applicants. When expatriates frequently return to their home countries, they can cause operational issues and cause local employees to feel insecure about the subsidiary. It takes time for newcomers to adjust to their new surroundings, get to know their co-workers, and learn about their profession. Additionally, expatriate shifts in learning and re-learning expenses are sources of inefficiency [21].

Another issue is the scarcity of qualified expatriates. International assignments have been affected by how sufficient the family support is for a sense of belonging [22]. Furthermore, the two most common reasons why expatriate applicants decline assignment offers are family considerations (37%) and their spouses’ jobs (19%) [23]. As a result, businesses are forced to fill openings with underqualified people [24]. For example, even though over 75,000 suitable managerial roles are required to be filled in China, only 3000–5000 managers will be available [25]. Consequently, MNCs can take advantage of local staff to address these aforementioned issues; however, MNCs are hesitant to pursue active staff localization due to an agency problem [26]. The agency problem is based on divergent interests and an information asymmetry [27]. To solve this problem, HQs monitor and verify their agents’ behaviours (i.e., local staff) [28], which leads to their demotivation to be engaged in the organizations. Thus, whether an employee is a local one or not can affect their intention to leave.

Although the importance of location to an MNC’s international operations has been widely debated, there has been little empirical research on the impact of localization on a subsidiary’s performance to date [29]. According to the studies on localization, MNCs hire local staff for various reasons. For example, MNCs benefit from localization because it allows them to tap into local expertise [27,28]. Moreover, legitimised MNCs may gain access to scarce local resources and information [29,30]. Localization aids the retention of capable local staff by cultivating their loyalty to a MNC [7]. However, few empirical studies have shown a link between management localization and success at the subsidiary and company levels. Furthermore, there have been only a few academic studies performed. To review previous studies regarding antecedents and outcomes of staff localization, we have summarized the findings in the Table 1.

### 2.2. Literature Reviews of Psychological Contract and Turnover

Organisational researchers have long used the notion of social exchange [46] and the norm of reciprocity [47] to describe the motivational basis for employee behaviour and the establishment of favourable employee attitudes (e.g., [48,49]). The psychological contract is an important framework to understand the employment relationship [50]. Employee perceptions of the relationship component of the psychological contract will impact their future attitudes and behaviours toward an organisation, including information sharing [51]. The fundamental concept of this agreement is to foster successful exchanges, particularly as they relate to employment. Similarly, if local staff believe that their MNCs are willing to delegate crucial roles to them, they are more likely to perceive a psychological contract with their possessions.

Previous studies have found that local employees are more likely to leave because they have fewer prospects for career progress [8,52]. This can be explained by social exchange theories, which hold that people form relationships with others to obtain the most out of them [46,53]. Furthermore, social exchange theories emphasise the need to comprehend employees’ motivations and how they relate to achieving corporate objectives [54]. The reciprocity norm governs social transaction relationships [47]. This theory helps explain why employees are motivated to carry out specific behaviours as a part of their mutual commitment to their employers.

The psychological contract, which can be defined as a set of unwritten expectations present at any given time between each member of an organisation and others in that organisation [46,47], is a construct derived from the social exchange theory and the norm of reciprocity [55]. According to the psychological contract literature, employees believe that the resources they have received are compelled to be provided to the organisation and that the organisation is obligated to offer them in return [56].

The core principle of this contract is that an understanding of the individual parties’ attitudes about their exchange relationships is critical to developing successful exchanges, especially in the context of employment [57]. In general, psychological contracts emphasise the importance of an employee’s discontent with poor performance as a primary determinant of the organisation’s perceived failure to maintain its commitments [54]. From the perspectives of local staff, expatriation could result in psychological contract violations and this is linked to a slew of harmful consequences [58].

### 2.3. Hypothesis Deveopment

#### 2.3.1. Staff Localization and Turnover

One of the most crucial challenges in American organisations is the “great resignation”, since there have been record-high voluntary turnover rates with more than 40% of workers considering quitting their positions [59]. Researchers and practitioners have worked to determine the reasons why workers leave their jobs and have offered suggestions for employee retention. Reduced job satisfaction has decreased organisational trust, increased turnover, decreased the feelings of obligation to one’s employer, reduced the willingness to participate in organisational citizenship behaviours, and decreased work performance, which are all likely outcomes of perceived psychological contract violations [60,61]. Although some research has been conducted to build and empirically evaluate the localization models, e.g., [21,62,63], the studies on retaining local staff have been conducted by comparing nationalities.

Over the last few decades, voluntary employee turnover has received considerable attention [52]. Although employee turnover has been linked to a poor organisational performance [64], the research has traditionally concentrated on what causes employees to leave rather than to stay, leaving the area of employee retention under-researched [65]. Local employees are the primary source of expertise and social networks for leveraging a firm; therefore, maintaining key local employees is vital for long-term international competitiveness [52].

As a voluntary phenomenon, employee turnover refers to an individual’s self-initiated and permanent termination of membership in a company [66]. There is a scarcity of research on the challenges of local employee turnover in a multinational setting, such as the existing pay disparities between expatriates and local employees [67] or the potentially negative consequences of international staffing practices for subsidiary employees [52].

According to the psychological contract theory, local staff who believe their MNCs are prepared to assign key duties to them are more likely to perceive a psychological contract with their sense of belonging, and staff in the subsidiary then feel obligated to supply the organization with the resources they have been given, and that the organization is obligated to provide them in return [56]. However, if there are a relatively high number of expatriates compared to other subsidiaries in an MNC, the local staff can perceive a low level of prospects in the relationship between his or her efforts and their sense of belonging. Local staff are likely to believe that expatriates play an important role although local nationalities put efforts into an organization. When the local employees perceive broken promises (i.e., a violation of a psychological contract), they might have various negative emotions, such as disappointment, frustration, a sense of betrayal and anger [59]. Based on this rationale, we developed the following hypothesis.

**Hypothesis** **1** **(H1).***The more local staff that retains compared to expatriates in the subsidiary, the less local employees leave their organization*.

#### 2.3.2. Moderating Effect of a Foreign Subsidiary CEO’s Nationality

Organizing the relationship between the headquarters and a subsidiary is one of the most important issues for MNCs [68]. To strike a balance between centralization and decentralization, the MNCs’ structures have evolved towards network-based systems [69]. To pursue decentralized MNCs, the role of their subsidiaries is important to contribute to the MNCs’ market development, and a subsidiary is a crucial source of bargaining strength [70]. The embeddedness of a subsidiary in a local market has been demonstrated to be highly linked to its importance to production or product development [71].

Subsidiaries take advantage of the local knowledge and embeddedness of their local executives [72]. Compared to expatriate CEOs, local CEOs are better positioned to guide the subsidiaries toward more market alignment through isomorphic adaptation [73]. They understand the local demand preferences [74] and have a higher level of embeddedness in local networks [75]. Furthermore, executives in the organisation play a vital role as signals to internal employees and external stakeholders. In particular, the recruiting practices and company policies send signals to employees [76]. Through signals from the executives to employees, the signal receivers may have positive or negative reactions and feedback [77].

When a subsidiary assigns local staff to an executive position, the local staff are more likely to receive strong signals that they could serve high level positions in the firm [29]. In other words, local staff have additional prospects for advancement inside a company [8]. Local staff are likely to have high turnover intentions if the subsidiary’s executive roles and other critical responsibilities are staffed by expats because promotion expectations and monetary compensation for the employees are not met from a psychological contract’s perspective [78,79]. The following hypotheses can be developed based on the psychological contract theory and previous studies.

**Hypothesis** **2** **(H2).***As the local staff takes on the CEO’s role in the subsidiary, the negative impact of staffing localization on turnover tends to be buffered. The overall research model is illustrated in*Figure 1.

## 3. Method

Our hypothesis testing required the estimation of turnover based on staff localization and a subsidiary CEO’s nationality by using a hierarchical linear regression [80]. To avoid a multicollinearity issue, we centred the variables at our means prior to calculating a term to estimate an interaction in the hierarchical linear regression [81].

### 3.1. Case Setting and Data Collection

We gathered data from 89 South Korean MNCs’ headquarters and subsidiaries in 25 countries across four regions for this study (America, Asia, Europe, and Oceania). We chose South Korea as the research setting for numerous reasons. First, some Korean MNCs are large, with a wide geographical reach in multiple subsidiary locations worldwide and with leading positions in their respective sectors [82]. Due to an ageing society and economic recession, South Korean MNCs seek to expand their businesses for internationalisation. Second, South Korea is a significant research topic that has attracted considerable interest in international business studies [83], and it is Asia’s fourth-largest economy and a strategically important market for several MNCs [84].

To test the hypotheses, the author sent a research proposal to global HR staff at 20 South Korean MNCs. Nine staff approved the proposal, and the author worked with them to seek information about the research. The question was issued to both HQs and subsidiary staff since the number of expatriates and the CEO’s nationality are managed by the HQs staff, but the turnover rate is maintained in the foreign subsidiary. Then, an email to the subsidiary was sent to the staff that the HQs staff selected, which was written in English, with an explanation of the research.

From September to December 2020, we collected the relevant data through the participating MNC’s HQs and subsidiaries, which resulted in 89 sets. The manufacturing industry, which employs 89 companies, accounted for 56.2% of the total population, followed by the service (30.3%) and food (13.5%) industries. The MNC subsidiaries in this study had an average of 13 years of operation and 505 local employees. There were 25 nations represented in our sample: China (19), the United States (11), Indonesia (7), Vietnam (7), Japan (5), Russia (5), Malaysia (4), Australia (3), Germany (3), India (3), Myanmar (3), Pakistan (3), New Zealand (3), Singapore (2), Uzbekistan (2), and Belgium, the Czech Republic, Hungary, Kazakhstan, Mexico, Philippines, Poland, Taiwan, Thailand, and Turkey (one each).

### 3.2. Measurement Instrument

The related variables were collected from staff in the HQs and subsidiaries through an internal archival dataset. The explanatory variable (staff localization), potential moderating variable (CEO nationality in a foreign subsidiary), and response variable (turnover rate) were all measured along with the demographic variables (subsidiary size). The independent variable (ratio of local managers in the subsidiary to the number of expatriates) was operationalised by the ratio: the number of local managers/(the number of local managers + expatriates) × 100. The CEO nationality was measured by the proxy 0 = local staffs or 1 = expatriates. These two measurements were provided by the HQs staff. The outcome variable (turnover rate) was measured through a Likert scale (where 1 = 0~10%, 2 = 11~15%, 3 = 16~20%, 4 = 21~25%, and 5 = more than 26%). The turnover rate was given by the subsidiary staff in other countries.

The demographic information of the subsidiary size was used as a control variable. The subsidiary size was evaluated using a Likert scale (were 1 = 1–30, 2 = 31–50, 3 = 51–100, 4 = 101–300, 5 = 301–500, and 6 = more than 500 employees). A firm’s size is associated with its turnover since larger organizations may have a lower turnover rate due to its sophisticated HR practices and greater internal labour market opportunities [85,86,87]. Thus, we included the subsidiary size as a control variable in the analysis.

### 3.3. Analytic Procedure

For the empirical study, we used the SPSS 22 software to examine the hypothesised linkages between the constructs using a hierarchical linear regression. Several scholars have used the hierarchical linear regression approach in international business [15,24,88]. Before conducting the analysis, we centred the variable to minimize the multicollinearity issue. Additionally, we visualized the simple slope to prove the moderating effect.

## 4. Results

Table 2 shows the descriptive statistics of the study variables (e.g., the mean, standard deviations, and correlation coefficients). The staff localization variable had a negative association with the turnover rate. The subsidiary size had a positive relationship with the staff localization and a negative relationship with the turnover rate.

In our turnover estimation model (Table 3), the effect of staff localization on the turnover rate was negative and statistically significant (*β* =−0.08, *p* < 0.05), indicating that if the subsidiary had a lower number of expatriates, the local staff were not likely to leave the subsidiary. In other words, when the subsidiary ran on local staff and the staffing strategy focus was on staff localization, the local employees might have perceived their career visibility in the future. Accordingly, hypothesis 1 was supported. Furthermore, hypothesis 2 suggested that the CEO nationality in a foreign subsidiary can buffer the relationship between staff localization and the turnover rate, which was supported (*β* = −0.13, *p* < 0.05).

As the subsidiary CEO was likely to be a local manager, the negative relationship between the staff localization and turnover became stronger (see Figure 2). To test the moderating effect, we followed Aiken and West’s [89] guidelines to investigate the simple slope. For example, if the interaction terms of the independent and moderating variables indicate a significant association, the graph of the interaction is recommended to be schematically drawn using the mean value and 1 standard deviation, and then a basic slope test should be performed [90]. A simple slope was computed to understand the form of the moderating impact [91].

## 5. Discussion

### 5.1. Summary of Results

Are there any other approaches to reduce the turnover rate of local staff besides a higher salary and decent welfare? Driven by this question, this study presented a staff localization theory and CEO staffing for lowering staff turnover. By incorporating the psychological contract theory, this study built a macro-level moderating framework to investigate how staff localization affects the turnover through a CEOs’ nationality. Additionally, it examined the tendency of staff localization to affect the turnover. Based on the data, we confirmed that staff localization was negatively related to a higher turnover rate in a subsidiary, and that the CEO’s nationality allows the relationship between the staff localization and turnover to be strong. This means that if a local manager plays the role of the CEO of a subsidiary, then the turnover rate can be reduced by interacting with staff localization.

### 5.2. Theoretical Implications

This study makes theoretical contributions to the academia by expanding our understanding of the impact of staff localization. The objective of this study was to investigate why local staff leave MNC subsidiaries based on a psychological contract [8]. Previous analysis has demonstrated that psychological contracts can be a foundation to understand the interpersonal dynamics within and outside organizations [92]. In particular, this study investigated the relationship between local staffing and turnover in different contexts, that included 25 countries through different data sources. In doing so, the study can enhance the understanding of different contextualized impacts on staff turnover from staff localization.

This study also adds to the body of knowledge on local staff turnovers. Given that the relationship between staff localization and turnover is negative [21], this includes whether a CEO is a local one as a moderator of that local staff turnover. Yang & Pak [8] resolved the complicated relationship between staff localization and a turnover intention. On top of their findings, this study adds value to the academia since the study used actual turnover rates rather than an intention to leave. The human resource management teams of MNCs need to pay attention to their staffing strategies for their subsidiary CEOs [93]. In doing so, this literature extends the understanding of CEO staffing in MNCs. Additionally, competent local staff may regard themselves as having fewer career advancement options and as being in competition with expatriates because the expatriates tend to assume managerial roles [52]. This study responds to the demand for research on the complex relationships between expatriates and local staff [94,95].

Finally, we used the psychological contract theory to explain the mechanism between staff localization and the turnover rate of a subsidiary [96], including the CEO’s nationality. In other words, the actual management positions in the subsidiaries, owing to the localization strategies that are critical for retaining local staff, may raise local staff expectations and enhance their positive attitudes, such as a long-term growth vision and development plan [21]. Whether local staff play the role as a CEO of a subsidiary, local staff can be engaged in their organizations since the CEO sends a strong signal to those local staff [8]. By confirming this impact in this study, we confirmed the CEO’s signalling impact on employees, especially in the MNC context.

### 5.3. Practical Implications

In a practical sense, these findings have significant implications for management. First, MNCs have a limited capacity to grow their enterprises into new nations due to a lack of competent employees [97]. This problem is especially significant in fast-growing emerging countries, such as Brazil, China, India, and in Eastern Europe [98]. Consequently, the issue of distributing a limited recruitment pool across several institutional contexts is relevant. Our findings imply that, although there is no one-size-fits-all pattern for MNCs [99], staff localization can lower the turnover of local staff in their subsidiaries.

Second, the CEO’s nationality also matters in the turnover of local staff. Previous studies have paid attention to the relationship between a CEO’s nationality and an organisation’s financial performance [100,101]. On top of that, the CEO’s nationality can influence individual-level outcomes including staff turnover. In that sense, this study emphasised the importance of staffing of a subsidiary’s CEO, since their nationality can influence a firm’s performance as well as individual-level outcomes; therefore, HR managers should be aware of this impact and consider a CEO’s nationality.

Third, digital transformation and the COVID-19 pandemic has changed the way expatriates and local staff collaborate. Expatriates are not likely to visit their subsidiaries in the way they did before the pandemic, due to travel prevention and digitalized tools such as Teams, Zoom, and others; therefore, the focus of staffing strategies in MNCs should shift from replacing expatriates to preparing well-prepared local staff. In the transition of usage of local staff in South Korean MNCs, this study provides reasons for why staff localization should be pursued by focusing on the turnover rate.

### 5.4. Limitations and Future Research Directions

Despite these contributions, this study had some limitations. First, although there are reasons why we chose South Korea as the research setting, it is better to have a variety of countries across the globe to increase the generalizability [102,103]; therefore, future studies can examine the impact that this study verifies in different contexts [104]. Second, the operationalisation of each variable needs to be strengthened because this study used a single item for each variable [105]. For example, compared with the actual turnover rate in a subsidiary, we could gather data on the turnover intention, which is a psychological response to staff localization. By doing so, we could then observe the impact of a psychological contract violation. Third, this study might have an endogeneity bias since the data was not collected from a randomized controlled experiment setting, thus leading to a reverse causality issue. As a result, we strengthened the theoretical background to explain the causality [106,107]. For future studies, it is highly recommended to consider the endogeneity issue when designing the research [108].

## 6. Conclusions

Taken together, staff localization can lower the turnover for subsidiary staff since they believe that a psychological contract is kept and can maintain their sense of belonging. If there is a relatively larger number of expatriates in a subsidiary, the local staff are likely to perceive a violation of their psychological contract. In the same sense, a CEO’s nationality can signal the visibility of career progression to the local employees. If expatriates fill a subsidiary’s CEO position, then local employees are not likely to believe that they will play a role as a leader of the subsidiary in the future. Therefore, a local staffing strategy focusing on increasing the number of local staff and assigning local employees in the CEO position can be important in managing a subsidiary’s local staff.

## Figures and Tables

**Figure 1 behavsci-12-00402-f001:**
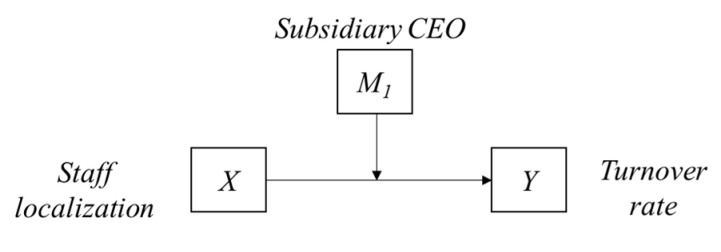
The theoretical research model (made by author, 2022) Research Model.

**Figure 2 behavsci-12-00402-f002:**
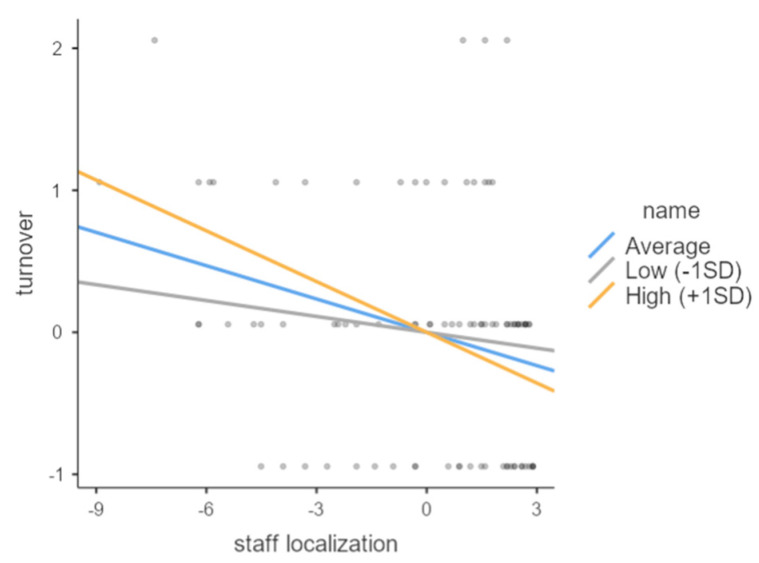
Moderating effect of subsidiary CEO on the relationship between staff localization and turnover rate.

**Table 1 behavsci-12-00402-t001:** Antecedents and outcomes of staff localization.

Antecedents	Outcomes
-Planning and selection of expatriate (Fryxell et al., 2004 [31]) -Willingness of expatriate to train local managers (Selmer, 2004a [32]) -Identified selection, recruitment and retention of suitable local employees (Selmer, 2004b [33]) -Local government, local managers and a firm’s parent company (Hailey, 1996 [34]) -Fears of growing unemployment and economic and demographic factors (Yamani, 2000 [35]) -Clear preference of restricting jobs for the locals (Sparrow et al., 2016 [36])	-Cost reduction, local market knowledge, and retention of local managers (Bjorkman et al., 1997 [37]) -Cost reduction, avoidance of expatriation drawbacks, a deepening of local knowledge, and development and retention of local managerial talents (Fayol-song, 2011 [7]) -Cost reduction by 50% (Hauser, 2003 [20]) -Average ratio of market value to book value (Moideenkutty et al., 2016 [38]) -Cost effectiveness (Banai and Harry, 2004 [39]; Scullion and Brewster, 2001 [40]; Black, 1999 [41]; Harzing, 1995 [42]) -High performance compared to expatriates (Harry and Collings [43], 2006; Gregersen and Black, 1992 [44]; Gamble, 2000 [45])

**Table 2 behavsci-12-00402-t002:** Means, Standard Deviations, and Correlations.

Variable	Mean	SD	1	2	3
1. Staff localization	97.1	2.93	-		
2. Turnover	2.19	0.98	−0.29 **	-	
3. CEO nationality	0.64	0.48	−0.19 *	0.24 *	-
4. Size	2.97	1.46	0.29 *	−0.19 *	−0.05

Note. N = 89. * *p* < 0.05, ** and *p* < 0.01 (two-tailed).

**Table 3 behavsci-12-00402-t003:** Results of hierarchical linear regression.

Variables	Turnover			
	Model 1	Model 2	Model 3	Model 4
Control variable				
Size	−0.131	−0.080	−0.086	−0.072
Independent variable				
Staff localization		−0.085 *	−0.075 *	−0.72 *
*Moderating variable*				
CEO’s nationality			0.483 *	0.510 **
Moderating variable				
Staff localization × CEO’s nationality				−0.132 *
ΔR^2^		0.058	0.051	0.030
R^2^	0.037	0.095	0.147	0.177
F	3.364 *	4.536 *	4.878 **	4.519 **

Note. N = 89. *p* < 0.10, * *p* < 0.05 and ** *p* < 0.01 (two-tailed). Unstandardized regression coefficients are shown. Interaction term is based on mean-centred scales.

## Data Availability

The data of this study are available from the corresponding author upon reasonable request.

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
