# Peer review of "Impact of Staff Localization on Turnover: The Role of a Foreign Subsidiary CEO"

_behavsci, 2022, doi:10.3390/bs12100402_

Round 1

Reviewer 1 Report

The subject is interesting and up to date, nonetheless, some areas need to be addressed:

Abstract

Further develop the abstract, and include at least 5 Keywords

1- Introduction

Line 26, change to Furthermore, also line 259

It is suggested to include a paragraph at the end of the Introduction describing how the paper is organized.

2- Theory and Hypotheses

This part is confusing, it is suggested to create a separate part for the literature review and another for the hypotheses (or bring it to the introduction part).

3- Method

At the beginning of this part, explain the methodology used, supported by actual literature.

5- Discussion and Implications

Results should be further discussed in this part, on the other hand, it is suggested to create part 6 (Conclusions), where besides the conclusions, the Theoretical Implications; Practical Implications; Limitations and Future Research Directions, and clues for future work could be addressed.

Author Response

We are excited that you see value in our manuscript (behavsci-1817029) and thank you for the opportunity to revise and resubmit our manuscript to the Behavioral Sciences. Below we have reproduced reviewers’ feedback in bold italics, followed by our response (blue color) in regular font. We carefully read the three reviewers’ comments and addressed and/or responded to each comment. We greatly appreciate this opportunity to revise and resubmit our manuscript. We welcome your continued feedback as we strive to make our manuscript as helpful to the field as possible.

Recommendation:

The subject is interesting and up to date, nonetheless, some areas need to be addressed

Comments:

  1. Abstract: Further develop the abstract, and include at least 5 Keywords

Response to reviewer: We greatly appreciate your suggestion on developing the abstract and including at least 5 keywords. First, we have included the environmental motivation of why local staff’s turnover is important by citing great resignation phenomenon, empirical findings in the abstract and one more keyword to reflect your feedback. Thank you for your helpful suggestion.

  1. Introduction: Line 26, change to Furthermore, also line 259. It is suggested to include a paragraph at the end of the Introduction describing how the paper is organized.

 Response to reviewer: Thanks for your comment. We have changed conjunction like you had mentioned (into furthermore). Also, we have included a paragraph in the last part of introduction to represent how the paper will be presented. Thanks for your developmental feedback.

  1. Theory and Hypotheses: This part is confusing, it is suggested to create a separate part for the literature review and another for the hypotheses (or bring it to the introduction part).

 Response to reviewer: We greatly appreciate your suggestion of articles to improve our study.

We have separated theory and hypotheses section into literature review and hypotheses development sections to avoid potential confusion. Thanks for your feedback.

  1. Method: At the beginning of this part, explain the methodology used, supported by actual literature.

 Response to reviewer: Thanks for your comment. We have included what methodology was used for the empirical analysis at the beginning of the method section supported by previous literatures. Thanks for your feedback.

  1. Discussion and Implications: Results should be further discussed in this part, on the other hand, it is suggested to create part 6 (Conclusions), where besides the conclusions, the Theoretical Implications; Practical Implications; Limitations and Future Research Directions, and clues for future work could be addressed.

Response to reviewer: Thanks for your comment. Based on your suggestion, we created conclusion section after discussion. Also, we developed summary of results, and practical implications. Once again, thanks for your helpful suggestion.

Reviewer 2 Report

There are three critical issues to address in this manuscript: errors in model specification, problems with PLS SEM, and support for hypotheses. In addition, there are several lesser issues.

Model Specification errors

1.     Although a bit unclear, it appears you have postulated that relationships between percentage of non-local v nonlocal managers affects dropout among managers, and that the relationship is moderated by whether the CEO is local or non-local There is a probable problem of endogeneity bias here: For example, the type of organization or the probability of advancement will likely affect local versus ex pat status of the CEO and turnover rate. Given the absence of measures of other variables that explain both the DV and the IV, , the effect of CEO status on turnover is inaccurate.

2.     This is essentially a mediated correlation analysis. SEM is premised on multiple independent variables to control for endogeneity bias. A simple model such as the one represented does not control for the major variables that can affect an outcome. Section 2.1 of your manuscript lists several variables that might mediate the relationship between the IV and DV, thus addressing exogeneity.  You need to cast a wider net in your lit review to identify possible sources of endogeneity bias.

3.     Your core premise is that the local/non-local status of managers affects turnover. You need to argue that this is a significant question and not the result is not self-evident.

PLS-SEM

Ronkko, McIntosh, Antonakis, and Edwards conducted an extensive analysis of PLS SEM in The Journal of Operations Management, 2016 (reference below). They found serious flaws is the procedure and recommended that it no longer be used in academic work. Some journals will now desk reject articles that use this research method. Unfortunately, your N is not sufficiently large to do a covariance SEM so this could be a big problem for you.  you might solve the problem by r revising the study as an HLM analysis. Even so, you still need to address endogeneity bias. I suggest that the journal editor decide whether PLS SEM will be acceptable for the journal. 

Model Support

Section 2.3 is intended to provide support for Hypothesis 2 but fails to do so. You need literature that specifically addresses moderation effects of CEOs on the relationship between local/non-local managers on turnover. 

Furthermore, section 2.3 drifts into network analysis without explanation. This is confusing and the findings reported do nothing to support the relationship identified in Hyp 2.

Other Issues

In section 2.3 you at times confuse CEOs and managers, making it hard to follow your logic.

You could help the reader by defining and explaining the organizational structure of multinational collaborations.

In section 3., line 233, I think you meant to refer to subsidiary size rather than age.

In section 2.2 you mention social exchange theory and appear to provide research from the LMX literature. You might go into these theories in a little depth—enough to explain the theories. 

Reference

Rönkkö, M., McIntosh, C.N., Antonakis, J., Edwards, J.R. (2016). Partial least squares path modeling: Time for some serious second thoughts.  Journal of Operations Management (47, 48), 9-27.

Author Response

We are excited that you see value in our manuscript (behavsci-1817029) and thank you for the opportunity to revise and resubmit our manuscript to the Behavioral Sciences. Below we have reproduced reviewers’ feedback in bold italics, followed by our response (blue color) in regular font. We carefully read the three reviewers’ comments and addressed and/or responded to each comment. We greatly appreciate this opportunity to revise and resubmit our manuscript. We welcome your continued feedback as we strive to make our manuscript as helpful to the field as possible.  

Recommendation:

There are three critical issues to address in this manuscript: errors in model specification, problems with PLS SEM, and support for hypotheses. In addition, there are several lesser issues.

Comments:

  1.  Although a bit unclear, it appears you have postulated that relationships between percentage of non-local v nonlocal managers affects dropout among managers, and that the relationship is moderated by whether the CEO is local or non-local. There is a probable problem of endogeneity bias here: For example, the type of organization or the probability of advancement will likely affect local versus expat status of the CEO and turnover rate. Given the absence of measures of other variables that explain both the DV and the IV, the effect of CEO status on turnover is inaccurate.

Response to reviewer: We greatly appreciate your feedback on endogeneity bias in the study. Like you mentioned, the current study has endogeneity issue especially affecting reversed causality issue. To address this issue, we have first investigated if there might be instrumental variable, but we failed to find it due to insufficient variables (we selectively collected the data from the internal HR system of target MNCs). Then, we have examined the paper of ‘From the Editors: Endogeneity in international business research’ in Journal of International Business Studies by Reeb, Sakakibara & Mahmood (2012) and sought to find the way to tackle the issue. Unfortunately, we could not find the proper method (e.g., natural experiments, regression discontinuity designs) but tried to explain the causality based on strengthening theoretical background like Reeb et al. (2012) emphasized in the last section of the paper. Also, we confessed the endogeneity issue of the current study in the limitation section and suggested authors to consider this issue for future study. Once again, many thanks for your insightful feedback.

  1. This is essentially a mediated correlation analysis. SEM is premised on multiple independent variables to control for endogeneity bias. A simple model such as the one represented does not control for the major variables that can affect an outcome. Section 2.1 of your manuscript lists several variables that might mediate the relationship between the IV and DV, thus addressing exogeneity.  You need to cast a wider net in your lit review to identify possible sources of endogeneity bias. 

Response to reviewer: We sincerely appreciate your suggestion on endogeneity bias. Based on your feedback and Rebb et al. (2012), we focused on strengthening literature review and tried to identify what variables might be related to IV or DV by summarizing the table of antecedents and outcomes. By doing so, we wish readers might find the variables that mediate the relationship between the IV and DV to address exogeneity issue for future study. Once again, thank you very much on the feedback.

  1. Your core premise is that the local/non-local status of managers affects turnover. You need to argue that this is a significant question and not the result is not self-evident. 

Response to reviewer: Thanks for your feedback. To reflect your suggestion, we have included why local staff might have the intention to leave the subsidiary by using agency problem. To address agency problem, HQs monitor the local staff behaviors, and it can function as demotivational factors for local managers, leading to lower level of engagement in the organization. In this logic, we argue that our question is important rather than self-evident one. We appreciate your helpful suggestion.

  1. Ronkko, McIntosh, Antonakis, and Edwards conducted an extensive analysis of PLS SEM in The Journal of Operations Management, 2016 (reference below). They found serious flaws is the procedure and recommended that it no longer be used in academic work. Some journals will now desk reject articles that use this research method. Unfortunately, your N is not sufficiently large to do a covariance SEM so this could be a big problem for you. you might solve the problem by revising the study as an HLM analysis. Even so, you still need to address endogeneity bias. I suggest that the journal editor decide whether PLS SEM will be acceptable for the journal.  

Response to reviewer: We greatly appreciate your suggestion. First, we have reviewed the Ronkko et al. (2016) paper and found that there should be an issue to use PLS SEM for the study. Then, we have retested our hypotheses by using hierarchical linear regression analysis through SPSS 22 software after centring the variable to avoid multicollinearity issue. Also, we visualized the simple slope to prove the moderating effect. For the future study, we will gather enough data and independent variables to use SEM to avoid these issues.

  1. Section 2.3 is intended to provide support for Hypothesis 2 but fails to do so. You need literature that specifically addresses moderation effects of CEOs on the relationship between local/non-local managers on turnover. Furthermore, section 2.3 drifts into network analysis without explanation. This is confusing and the findings reported do nothing to support the relationship identified in Hyp 2.

Response to reviewer: Thanks for your comment. First, we are really sorry to confuse you since we have reverse-coded the CEO’s nationality to make the negative direction like the direct effect to avoid confusion for readers. We should have changed the proxy of CEO’s nationality the other way around. Thanks to your correction, we have made a change in the measurement section. Also, like you mentioned, we have revised section 2.3 with the explanation of embeddedness and decentralization of the subsidiary to minimize the confusion for readers. Also, we have changed our hypotheses 2 since it appears confusing to readers. Once again, thanks for your developmental feedback.

  1. In section 2.3 you at times confuse CEOs and managers, making it hard to follow your logic. You could help the reader by defining and explaining the organizational structure of multinational collaborations.

Response to reviewer: Thanks for your comment. Based on your suggestion, we have changed the term ‘manager’ into ‘staff(s)’ throughout the paper. We intended to describe the employee in the subsidiary as either manager or employee, but it confused the readers like you pointed out. Thanks.

  1. In section 3., line 233, I think you meant to refer to subsidiary size rather than age.

Response to reviewer: Thanks for your comment. I have corrected the word from age into size.

  1. In section 2.2 you mention social exchange theory and appear to provide research from the LMX literature. You might go into these theories in a little depth—enough to explain the theories. 

Response to reviewer: Thanks for your comment. Based on your suggestion, we have included the literatures related to violation of psychological contract since we have focused on the break of psychological contract. Once again, many thanks.

Reviewer 3 Report

This paper investigates the potential connection between localised managerial leadership and staff turnover in large multi-national corporations. This is a very interesting paper that is generally well written. The strength of the paper is that this is an interesting topic of study, particularly from a managerial perspective in terms of its scope and impact. However the paper needs improvement by addressing a the three issues outlined below.

1. The first issue is that there needs to be a clearer articulation of the aim of the paper at the outset, in the introduction, and this needs to be consistent throughout. The abstract contains the following aim:

“This study examines the macro-level relationship between the ratio of local managers in the subsidiary and the actual turnover rate of 89 multinational companies in 25 countries through headquarters and subsidiary managers.”

The paper fails to articulate the aim of the paper until the discussion section, where it implied the aim slightly differently:

“… this study built a macro-level moderating framework to investigate how staff localization affects turnover and CEOs’ nationality moderates the relationship between staff localization and turnover.”

The aim expressed in the abstract suggests that the aim of the paper is to merely examine the “ratio” of local managers to staff turnover, whereas the discussion section suggests that the aim of the study is to examine “how” localization “affects” turnover and nationality moderates the relationship between staff. Does the paper really examine “how” these variables affect one another? Or is it really a case of seeking correlations between variables that maybe warrant greater attention afterwards?

2. The second issue is connected to the previous point: there needs to be a stronger justification for the study that motivates the reader. This needs to be placed in the introduction. It is not adequate to simply state that this issue has not previously been examined. Perhaps there is a very good reason for a lack of research in this area. The key point is to offer a much stronger argument for why we need this study.

3. The third issue is that the theoretical contribution is not fully articulated and needs to be clearly articulated. Currently the claimed “theoretical contributions” are not presented as theory contributions because they lack explanation. The contributions are claimed to “expanding our understanding of …” and that they “add to our understanding of …” yet the paper presents very little evidence of explaining any of these things. To improve the paper you need to demonstrate how these findings expand upon what we previously knew by giving the reader depth of insight and, in particular, how this relates to the previously cited papers. Whilst there may be a correlation between nationality of leaders and staff turnover, there could also be alternative explanations.

Currently the contributions presented in the paper might call attention to the potential impact of leadership on staff turnover but they do not show how these findings build upon earlier studies adequately.

Author Response

Please provide a point-by-point response to the reviewer’s comments and either enter it in the box below or upload it as a Word/PDF file.

: We are excited that you see value in our manuscript (behavsci-1817029) and thank you for the opportunity to revise and resubmit our manuscript to the Behavioral Sciences. Below we have reproduced reviewers’ feedback in bold italics, followed by our response (blue color) in regular font. We carefully read the three reviewers’ comments and addressed and/or responded to each comment. We greatly appreciate this opportunity to revise and resubmit our manuscript. We welcome your continued feedback as we strive to make our manuscript as helpful to the field as possible. 

Reviewer: 3

Comments:

This paper investigates the potential connection between localised managerial leadership and staff turnover in large multi-national corporations. This is a very interesting paper that is generally well written. The strength of the paper is that this is an interesting topic of study, particularly from a managerial perspective in terms of its scope and impact. However the paper needs improvement by addressing a the three issues outlined below.

  1. The first issue is that there needs to be a clearer articulation of the aim of the paper at the outset, in the introduction, and this needs to be consistent throughout. The abstract contains the following aim:

“This study examines the macro-level relationship between the ratio of local managers in the subsidiary and the actual turnover rate of 89 multinational companies in 25 countries through headquarters and subsidiary managers.”

The paper fails to articulate the aim of the paper until the discussion section, where it implied the aim slightly differently:

“… this study built a macro-level moderating framework to investigate how staff localization affects turnover and CEOs’ nationality moderates the relationship between staff localization and turnover.”

The aim expressed in the abstract suggests that the aim of the paper is to merely examine the “ratio” of local managers to staff turnover, whereas the discussion section suggests that the aim of the study is to examine “how” localization “affects” turnover and nationality moderates the relationship between staff. Does the paper really examine “how” these variables affect one another? Or is it really a case of seeking correlations between variables that maybe warrant greater attention afterwards?

Response to reviewer: We greatly appreciate your feedback on the aim of the study. Thanks to your comment, we make the aim of the study clearer by including it in the introduction and the discussion section. First, we investigate the relationship between staff localization and the actual turnover since the further study is needed according to previous studies. Second, the mechanism of how staff localization affects the turnover requires more studies. In that sense, we emphasized the aim of the study in each section. Once again, thank you for your helpful suggestion.

  1. 2. The second issue is connected to the previous point: there needs to be a stronger justification for the study that motivates the reader. This needs to be placed in the introduction. It is not adequate to simply state that this issue has not previously been examined. Perhaps there is a very good reason for a lack of research in this area. The key point is to offer a much stronger argument for why we need this study.

Response to reviewer: Thanks for your insightful comment. First, we included the reason why turnover issue is important by citing great resignation phenomenon in the abstract and the staff localization section. Second, we emphasized the reason why there was a lack of previous studies on local staff’s turnover rate by including the difficulty to gather the actual data of the ratio of local staff and turnover rate. In fact, although turnover rate is an important information to manage for scholar to figure out the reason, it is hard to gather the information since the number is a hidden information. In that sense, this paper found the motivation for the research in international business discipline. Thanks for your developmental feedback.

  1. The third issue is that the theoretical contribution is not fully articulated and needs to be clearly articulated. Currently the claimed “theoretical contributions” are not presented as theory contributions because they lack explanation. The contributions are claimed to “expanding our understanding of …” and that they “add to our understanding of …” yet the paper presents very little evidence of explaining any of these things. To improve the paper you need to demonstrate how these findings expand upon what we previously knew by giving the reader depth of insight and, in particular, how this relates to the previously cited papers. Whilst there may be a correlation between nationality of leaders and staff turnover, there could also be alternative explanations.

Currently the contributions presented in the paper might call attention to the potential impact of leadership on staff turnover but they do not show how these findings build upon earlier studies adequately.

Response to reviewer: Thanks for your feedback. We emphasized the importance of this study in terms of resolving the complicated relationship between staff localization and turnover aligned with Yang & Pak (2022) study. Also, the study confirmed the CEO’s signaling impact on the employees, especially in MNC context. This can make a contribution to the literatures (Vachani, 2005; Kaur & Singh, 2019). Third, the finding of the study can be beneficial to the HQs and subsidiaries since they struggle to manage expatriates and local staffs to respond the change in a way of work due to COVID-19 and digital transformation. In that sense, more studies need to be conduct and this study can contribute to the professionals in international human resource management. Thanks for your helpful suggestion.

Reviewer 4 Report

I enclose the article with some suggestions for improvement.

Author Response

We are excited that you see value in our manuscript (behavsci-1817029) and thank you for the opportunity to revise and resubmit our manuscript to the Behavioral Sciences. Below we have reproduced reviewers’ feedback in bold italics, followed by our response (blue color) in regular font. We carefully read the three reviewers’ comments and addressed and/or responded to each comment. We greatly appreciate this opportunity to revise and resubmit our manuscript. We welcome your continued feedback as we strive to make our manuscript as helpful to the field as possible

Comments:

  1. Improvement of English readability.

Response to reviewer: We greatly appreciate your feedback on improvement of English readability. We have checked the grammar and spelling like you had suggested. Thank you for your helpful suggestion.

  1. Improvement theoretical framework: The theoretical framework does not offer enough arguments for your first hypothesis development.

Response to reviewer: Thanks for your comment. To reflect your feedback, we have included the content of violation of psychological contract and how it affected local managers. We have summarized previous literatures and provided the table of how staff localization has been studied. Also, we have changed the hypothesis 1 statement from ‘staff localization is negatively related to the turnover rate in the subsidiary’ to the more local staff exists compared to expatriates in the subsidiary, the less local employees leave their organizations’. We appreciate your suggestion.

  1. Please describe broadly how the hypothesis H1: Staff localization is negatively related to the turnover rate in the subsidiary is supported by your research. It is not clear how you came to the conslusion that the first hypothesis is supported.

Response to reviewer: Thanks for your feedback. We have included how staff localization can be negatively related to low turnover rate in the subsidiary in the results section. Based on the psychological contract, local employees believe that they might want clear career visibility. When the headquarters of the MNCs deploy more expatriates, however, local managers might not be able to perceive they will play a role as a leader or decision-maker. In this sense, we argue that staff localization is negatively associated to low turnover rate. Once again, thanks for your feedback

  1. Conclusion: I miss the conclusion after this sentence:

Response to reviewer: Thanks for your comment. Based on your suggestion, we created conclusion section after discussion. Also, we developed summary of results, and practical implications. Once again, thanks for your helpful suggestion.

  1. Citation: Correction of references

Response to reviewer: Thanks for your feedback. We have corrected the list of references by checking the paper and adding DOI information.

Round 2

Reviewer 2 Report

Still exhibits an issue of exogeneity  but

otherwise improved